# Contactless probing of polycrystalline methane hydrate at pore scale suggests weaker tensile properties than thought

Dyhia Atig [1], Daniel Broseta [1], Jean-Michel Pereira [2] & Ross Brown [3]✉

Methane hydrate is widely distributed in the pores of marine sediments or permafrost soils, contributing to their mechanical properties. Yet the tensile properties of the hydrate at pore scales remain almost completely unknown, notably the influence of grain size on its own cohesion. Here we grow thin films of the hydrate in glass capillaries. Using a novel, contactless thermal method to apply stress, and video microscopy to observe the strain, we estimate the tensile elastic modulus and strength. Ductile and brittle characteristics are both found, dependent on sample thickness and texture, which are controlled by supercooling with respect to the dissociation temperature and by ageing. Relating the data to the literature suggests the cohesive strength of methane hydrate was so far significantly overestimated.

[1] CNRS/ TOTAL/ UNIV PAU & PAYS ADOUR E2S UPPA, Laboratoire des fluides complexes et de leurs réservoirs, UMR5150, 64000 Pau, France. [2] Navier, Ecole des Ponts, Univ Gustave Eiffel, CNRS, Marne-la-Vallée, France. [3] CNRS/ TOTAL/ UNIV PAU & PAYS ADOUR E2S UPPA, Institut des sciences analytiques et de physico-chimie pour l'environnement et les matériaux, UMR5254, 64000 Pau, France. ✉email: ross.brown@univ-pau.fr

Gas hydrates are crystalline solids in which small guest species—mostly gaseous at ambient conditions—are enclathrated in molecular sized cavities in an ice-like matrix, at concentrations typically ≈10–100 times higher than in liquid water. Gas hydrates are therefore stable at low temperature or high pressure, e.g. hydrates of natural gas (methane) are widespread in permafrost or in ocean sediments. Gigatonne deposits around the continental margins[1] are a long viewed potential source of energy[2,3].

Gas hydrates contribute to the stability of sea floor sediments, by their own cohesion or by cementing mineral particles[4]. However, they may dissociate when the pressure drops[5] or the temperature rises[6], or dissolve under increased salinity[7] or depletion of dissolved methane[8], for example by sulphate-reducing microbial consortia[9] or by ocean circulation[10]. The strength or weakness of gas hydrates may regulate methane seepage from deep geological reservoirs[5]. Evidence of the coincidence of failure of submarine slopes and of hydrate deposits is widespread worldwide[11]. Sea-floor failure implicating gas hydrates, whether by causes natural or on purpose to extract gas, could threaten off-shore installations[8,12,13] or contribute via several mechanisms to submarine landslides[14]. Shells of gas hydrate around gas bubbles[15–17] may influence dissolving in the water column and eventual release to the atmosphere. Understanding the mechanical properties of methane hydrate thus remains important for a variety of processes like production of natural gas by depressurization or heating[2], interpretation of seismic data, or evaluation of the likelihood of geophysical hazards.

The mechanical properties of such complicated systems as hydrate bearing sediments depend on many factors such as the properties of the hydrate itself, those of the sediment, hydrate pore habit, and adhesion between the hydrate and sediment vs. cohesion of the hydrate[18]. But reconciling high pressures and low temperatures with the measurement of small stresses and deformations is difficult, e.g. under the optical microscope. Not surprisingly, our appreciation of the behaviour of gas hydrates themselves is therefore very deficient, specially at pore scale. The difficulty of recovery and study of samples of hydrate-bearing sediments in in situ conditions, and the caveats of more or less realistic laboratory synthesis[18]-size and timescale among many others, led to various modelling approaches relating the mechanical behaviour of the composite sediment to the growth, pore habit and micro-mechanical properties of gas hydrates. These models range in scale from effective medium and homogenization approaches[19,20] through continuum mechanics finite elements[21,22], to molecular modelling of the hydrate itself, e.g. classical molecular dynamics[23–25] to ab initio studies[24,26]. But modelling would benefit too from more experimental checks on methane hydrate at pore scale.

Grain size strongly influences the mechanical properties of polycrystalline solids[27–30] and is expected to influence those of gas hydrates[31,32]. In 2007, ref. [33] compared grain sizes in marine deposits and in synthetic samples of methane hydrate and noted the likely impact of grain size on mechanical properties, but to our knowledge there are still no data on how grain size may influence them. A gap of several orders of magnitude in grain size and tensile strength separates simulations of polycrystalline methane hydrate in molecular dynamics simulations at scale 10–50 nm[24], and the data on grainy ice at mm scale[34], which are moreover of questionable pertinence to methane hydrate[21]. Ref. [32] reported the tensile strength of mm-sized samples of methane, carbon dioxide and tetrahydrofuran hydrates, but not grain sizes. Finally, we are not aware of simple traction tests of the elasto-plastic properties or the elastic modulus of methane hydrate at any scale.

This paper probes the tensile properties of polycrystalline methane hydrate at micron scale, by a novel method: a contact-less, thermo-induced stress is applied to a tenuous shell of hydrate grown in a thin glass capillary under a microscope. We correlate grain size with the elasto-plastic properties of the shell under monotonic or cyclic regimes of loading up to failure, and derive the elastic constant and tensile strength as functions of temperature and duration of annealing, which control the grain size. Combining the present results with what data are available in the literature, we suggest that the tensile strength of methane hydrate in marine or geological settings may be significantly less than currently supposed.

## Results

**Stress testing a thin shell of methane hydrate.** A bubble-free column of water is introduced against the closed end of a glass capillary (see Methods and Supplementary Note 1 for details). The capillary is mounted under an optical microscope and the water is equilibrated with methane at 15 MPa and room temperature ($T_r ≈ 20$ °C). The gas pressure remains constant thereafter. We quench the sample until nucleation of methane hydrate at temperature $T_n ≈ -23$ °C. In less than seconds, a thin polycrystalline cap of hydrate covers the meniscus. Figure 1 and Supplementary Video 1 illustrate subsequent events. Also within seconds, crystallites a few microns in size nucleate in the water behind the cap, while elongated needles and dendritic crystals appear further back in the water column, Supplementary Fig. 9. The consequent increase of pressure behind the cap expels a film of water over the glass, rapidly forming a halo[35]. The halo is here a microns-thick, cylindrical layer of hydrate, not in contact with the glass but riding as an inner sleeve between the gas and the water film. Its polycrystalline texture is similar to that of the cap. Like some salt deposits[36], juvenile polycrystalline methane[37] and other hydrates[38] are porous, providing here for slow coarsening and thickening. Polyhedral crystals up to ≈50 μm across emerge at long times from areas of smaller crystallites, mostly organized in one or two layers. Below we call the cap and halo collectively the hydrate shell, separating the water from the gas by an at first weakly permeable but slowly thickening barrier.

On visually detecting the hydrate, we raise the temperature to an annealing temperature, $T_a > T_n$, still well below the dissociation temperature, $T_{eq} = 16.3$ °C at 15 MPa. The supercooling during annealing, $\Delta T = T_{eq} - T_a$, is the first parameter controlling the texture and strength of the shell. The second is the annealing time, $t_a$, prolonged here up to 7 h after nucleation.

The halo spreads down the capillary at ≈1 μms$^{-1}$, to a final length of ≈1 mm. Tracers, such as occasional hydrate debris from behind the cap or fluorescent nano-beads in the water in some experiments, confirm that growth is fed by water flowing between the halo and the glass. The tip of the halo eventually stalls and adheres to the capillary wall, usually after 2–3 h. But the halo continues to thicken and lengthen to the left in Fig. 1c, as diffusion of water into the grain boundaries feeds crystal growth all down its length (see below). The halo thus pushes the cap, that like a leaky piston exudes water in a self-sustaining process, that is eventually stifled below detection by infilling of the porosity.

Considering the capillary as a model pore, we identify the halo with the grain- or mineral-coating pore habit of gas hydrates[39,40], in which a few microns-thick layer of water is sandwiched between the equally thin hydrate shell and the substrate, in samples grown in the laboratory in presence of free gas[41]. Hydrate halos are frequently observed in high resolution computed X-ray tomography[39,42]. They were first noticed relaying the nucleation of the hydrate between droplets of water

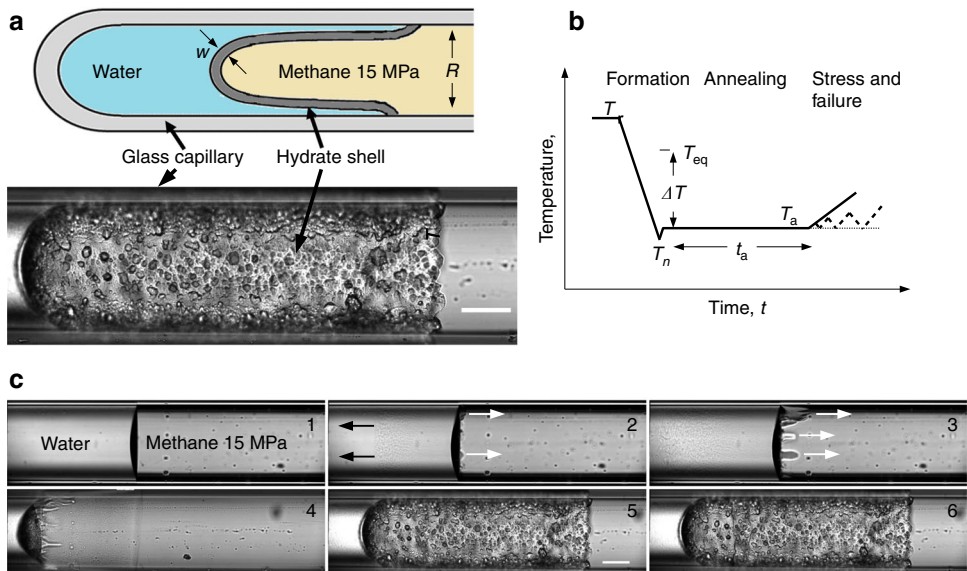

**Fig. 1 Principal steps of stressing a microns thick methane hydrate sleeve. a** Schematic and a typical sample in a glass capillary. A thin shell of methane hydrate (grey) separating water (blue) from the guest gas (yellow, pressure constant) is grown from the meniscus in a closed glass capillary, until it anchors to the glass. Scale bar 100 μm; **b** Steps in the formation and stressing of the hydrate shell. Methane-saturated water ($\approx$15 MPa, room temperature, $T_r$) is quenched until nucleation of the hydrate as a cap on the meniscus (here $T_n \approx -23\,°C$), initiating crystallization of micro-crystals in the water (black arrows in **c**) and a polycrystalline halo growing to the right on a film of water (white arrows). After annealing (hours) at temperature $T_a \geq T_n$, the sample is warmed ($\approx$1 K, at $\leq$1 K/min), or cycled to increasing temperatures (dashed line). Thermally induced contraction of the water causes tensile stress of the halo, who's elongation is measured under a microscope; **c** Snapshots of the process: 1: initial state; 2: nucleation of the cap, crystallites and halo; 3 and 4: spreading of the halo; 5 a late stage of annealing; 6: moment of failure at the serrated line near the former meniscus.

on flat glass[35]. They appear to be present in capillaries in refs. [43,44] and ref. [45] described in detail their growth in round capillary tubes, while ref. [46] adds qualifications to the case of flat glass. Here, the shell divides the capillary bore into water and gas compartments, with a degree of diminishing inter-compartment diffusion, that can be neglected or corrected on the much shorter timescale of the tensile tests below ($\approx$100 s), see Fig. 2b.

After annealing, we put the shell under tensile stress by slightly increasing the temperature at constant gas pressure, say a temperature excursion $\delta T(t)$ at time $t$ with respect to $T_a$. Since $T_a$ is below the inversion temperature of water, raising the temperature causes contraction, hence depression in the closed vessel formed by the glass and the hydrate shell, and tensile stress in the halo, reproducibly yielding remarkable circular fractures, perpendicular to the capillary axis. Let $\delta p = p_g - p_w$ be the thermo-induced pressure difference between the gas and the water sides of the shell. Viewing the halo as a thin-walled pipe with internal and external radii $r_i$ and $r_e$, and thickness $w = r_e - r_i \ll R$, the radius of the capillary bore, the induced axial stress is of order[47]:

$$\delta\sigma_a \approx \frac{R}{2w}\delta p \quad , \tag{1}$$

Let $\alpha_w$ and $\beta_w$ be the isobaric coefficient of thermal expansion and the isothermal compressibility of the aqueous phase, that to a first approximation undergoes an isochoric process. Then the change in pressure $\delta p(t)$, corresponding to a small temperature excursion $\delta T(t)$ from $T_a$, is deduced from:

$$\delta V/V = \frac{1}{V}\frac{\partial V}{\partial p}\delta p + \frac{1}{V}\frac{\partial V}{\partial T}\delta T = -\beta_w\delta p + \alpha_w\delta T = 0 \quad , \tag{2}$$

whence $\delta p(t) = \alpha_w\delta T(t)/\beta_w$. Using Eq. (1), the induced axial stress is estimated as

$$\delta\sigma_a(t) \approx \frac{\alpha_w}{\beta_w}\frac{R}{2w}\delta T(t) \quad . \tag{3}$$

The high thermal diffusivity and small size of the sample ($\approx$100 μm) ensure that its temperature and hence the water pressure closely track their target values. This work uses methane-saturated water, but study of e.g. brines would be relevant to gas hydrates. Fortunately, the values of $\alpha_w$ and $\beta_w$ of a particular aqueous phase may be measured in situ at the annealing temperature, by tracking the meniscus before formation of the hydrate, see Supplementary Note 3. The note also discusses small corrections to Eq. (3), together $\approx$10 %, that are included in the results below to account for the elasticity of the glass and its thermal expansion.

Because of the non-standard setup and method for traction tests, and because the mechanisms of growth of the halo are complex[48], we mention a few concerns. First is the complete stress state of the halo, initially in equilibrium with water and the gas at pressure $p_g$. Tensile testing might carry the internal pressure in the halo below the equilibrium pressure, causing dissociation[21]. The stress state at pressure differential $\delta p$ has radial ($r$), axial ($a$) and hoop ($h$) components, in order ($r_i \leq r \leq r_e$)[47]:

$$-p_g \leq \sigma_r(r) = -p_g + \frac{\delta p}{w}\left(\frac{r_e}{r}\right)^2\frac{r^2 - r_i^2}{r_e + r_i} \leq -p_w,$$

$$\sigma_a(r) = -p_g + \frac{\delta p}{w}\frac{r_e^2}{r_e + r_i} \approx -p_g + \delta p\frac{R}{2w}, \tag{4}$$

$$\sigma_h(r) = -p_g + \frac{\delta p}{w}\left(\frac{r_e}{r}\right)^2\frac{r^2 + r_i^2}{r_e + r_i} \approx -p_g + \delta p\frac{R}{w}.$$

The three-dimensional, thermally induced loading differs from the stress state in a conventional uniaxial tensile test. In order to compare our tensile failure data to available tensile strengths, we compute the equivalent von Mises stress, $\sigma_{eq}$,

$$\delta\sigma_{eq} = \sqrt{(3/2)s_{ij}s_{ij}} \approx \frac{\sqrt{3}}{2}\delta p\frac{R}{w}, \tag{5}$$

where $\mathbf{s} = \boldsymbol{\sigma} - \sigma_m\mathbf{1}$ is the stress deviator tensor, with $\sigma_m \approx -p_g + \delta p R/(2w)$ the average stress. The mechanical advantage

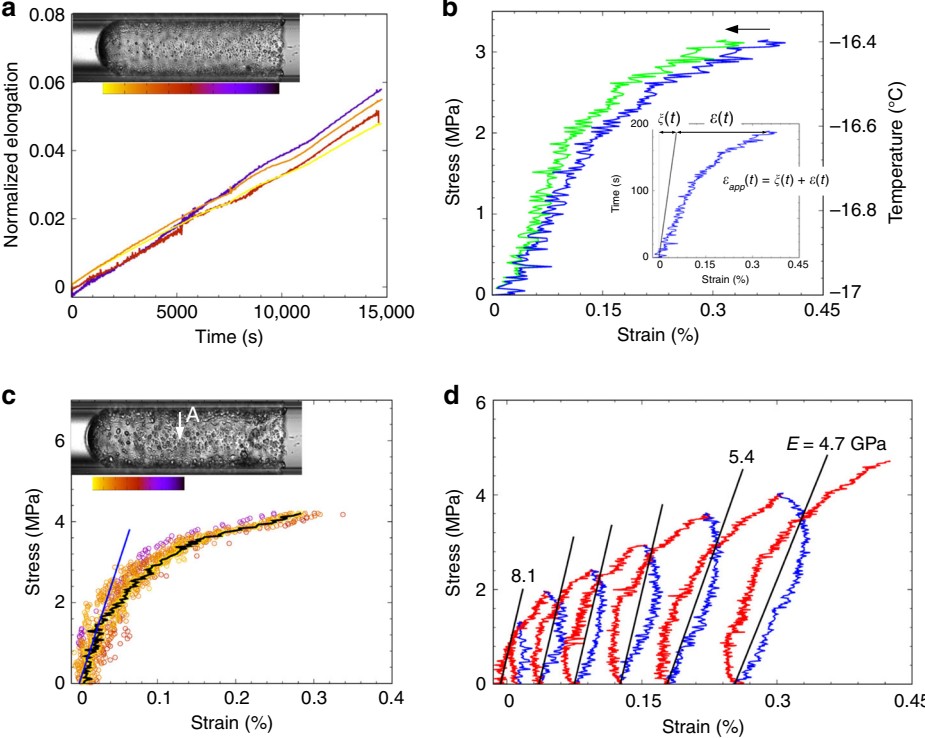

**Fig. 2 Axial stress *vs*. axial strain. a** Homogeneous crystal growth down the length of the sleeve during annealing, shown by the collapse to a common master curve, of the normalized deformation, $\xi(t)$ (Eq. (6)) at four locations with positions colour coded by the inset; **b** Application of a temperature ramp after annealing stresses the hydrate sleeve. Inset: correction of the apparent strain $\epsilon_{app}(t)$ (blue) for continuing crystal growth $\xi(t)$ to yield the stress-induced strain, $\epsilon(t)$ (green) ; **c** Stress–strain data for 13 sample points down the length of the sleeve (colour code in the inset), illustrate the homogeneous strain field under monotonic loading up to failure. Black curve : for point A in the inset; blue line : elastic modulus 5.9 GPa; **d** A cyclic tensile test shows plastic deformation, with increasing residual strains, and fatigue: the elastic modulus determined according to ref. [54], decreases with successive cycles. Data are for $t_a = 7$ h in **b–d** and for $\Delta T = 40.3$ K except (**b**), $\Delta T = 33.8$ K. Axial loading/unloading rates in **d** $\approx 4.5$ MPa/min (0.5 K/min, red branches), resp. $\approx -8.9$ MPa/min ($-1.0$ K/min, blue).

provided by the projected area of the cap *vs*. cross-section of the halo means that the pressure differential required to rupture the halo is small, $\delta p < 1$ MPa. Coupled with the thinness of the halo, this ensures that all stress components remain within the domain of stability of the hydrate, see Supplementary Fig. 8. Thinning under tension is too small to detect, making the Poisson ratio, $\nu$, inaccessible in this study. Therefore we report the measured axial elastic modulus, $E = \delta\sigma_a(t)/\epsilon(t)$, where $\epsilon(t)$ is the axial strain at time $t$, deduced from the video recordings (see "Methods" and Supplementary Note 2). It is related to Young's modulus, $Y$, by $E = \sqrt{3}Y/(1 - 2\nu)$.

Ice is a second concern. Despite the presence of strongly supercooled water, abandon of samples because ice nucleates before the hydrate is rarely necessary. Ice in those samples is unmistakable- fast (<1 s) conversion of the whole water column to an opalescent mass containing bubbles of methane. No evidence of ice is detected in the present data, e.g. changes on warming samples through the melting point of ice at the end of the runs.

The reason why water between the spreading halo and the glass does not produce hydrate directly is its depletion in methane by the initial formation of hydrate crystals behind the cap. The experimental configuration is very close to models of the spreading of hydrate films at the guest-water interface[49] or of film thickening[17]. Here, the concentration of dissolved methane must be close to equilibrium with the hydrate, $c \approx c_{hw}$. Spreading is fed by gas dissolving into grain boundries and into the meniscus stretched between the glass and the growing tip, at a

local concentration determined by the gas-water equilibrium, $c \approx c_{gw} > c_{hw}$ and diffusing down the concentration gradient to the halo tip cf. refs. [44,49]. Condensation in porous media[50], e.g. a breath figure of water droplets on the pore walls, might favour spreading of the hydrate in some excess gas situations, but was not detected here, contrary to our earlier work on halos of cyclopentane hydrate on strongly hydrophilic glass[51]. In that work, a breath figure of individual water droplets on the substrate gave rise to hydrate ridges parallel to the direction of spreading[51]. Here, micro-crystallites in the thin, juvenile halo are organized perpendicular to the line of growth, in ripples that might be due to slip-stick motion of the contact line, see Supplementary Fig. 10b.

Finally, the temperature ramps used to stress the shell could induce unwanted stress *via* thermal expansion, or could influence the crystal structure. The excursions are small, $\delta T(t) = 0.5$–2.5 K, ensuring that thermal expansion of the hydrate[52] and the glass contribute negligibly to the stress and to the measured strain. The expected crystal structure is sI throughout a run[53].

In summary, after annealing, we apply a small temperature ramp to exert traction on the halo *via* the thermally induced pressure drop in the water. So the stress rate is controllable and constant, rather than as more usual, the strain rate. The instantaneous deformation of the hydrate shell all down its length is provided by video-microscopy and image analysis. Thus we have both the stress and the strain ingredients of a contactless tensile test.

## Elasto-plastic properties of the hydrate shell

Minutes after the arrest of the halo tip during annealing, it becomes apparent that despite exclusion of stress by the constant temperature and pressure, the halo body is elongating to the left in Fig. 1c. We attribute the elongation to crystal generation at grain boundaries in the hydrate shell. Tracking the position $z(t)$ of features of the halo like crystallites ("Methods" and Supplementary Note 2), shows that their displacements are proportional to the distance from the immobile tip: the normalized rate of axial elongation,

$$\xi(t) = \frac{z(t) - z(0)}{z(0) - z_{tip}} \quad , \qquad (6)$$

is the same all down the halo, Fig. 2a, meaning that crystal growth at a given time is homogeneous. Therefore, the rate of growth is limited more by diffusion of gas or water across the halo than by the rate of seepage of feed-water between the halo and the glass. Similar results hold at supercoolings of 20–40 K, with the rate of elongation due to crystallization too weakly dependent on supercooling to determine a trend, $d\xi/dt \approx 3 \times 10^{-6}\,\text{s}^{-1}$. Because the tensile tests are short, we neglect any influence of stress (micro-cracks) on the rate of crystal growth and extract the true, stress-induced strain $\epsilon(t)$ from the apparent strain, $\epsilon_{\text{app}}(t)$ as, cf. Fig. 2b:

$$\epsilon(t) = \epsilon_{\text{app}}(t) - \xi(t) \quad . \qquad (7)$$

Monotonic tensile tests were conducted on hydrate shells annealed for 7 h at supercoolings 40.3, 33.8 and 21.8 K (Table 1). For clarity, Figs. 2c and 3a, b show only one test at given conditions. The instantaneous strain field is uniform, with linear and non-linear regimes preceding failure, Fig. 2c. Table 1 reports average elastic moduli and tensile strengths that increase with the supercooling. However, cyclic loading and unloading are better to probe the elastic domain. Here, we slightly warm and cool the sample in a sequence of temperature ramps of increasing amplitude, Fig. 2d. The residual strains at the end of each cycle are characteristic of increasing irreversible plastic strain. The elastic constant determined according to ref. [54] decreases during cycling, showing weakening of the hydrate shell.

## Supercooling, ageing, grain size and tensile properties.

The shell is similar to the hydrate skin on methane bubbles rising in sea-water, described in ref. [55]. Similarly to cyclopentane hydrate halos in the same experimental configuration[44], and to gas hydrate crusts at water-guest interfaces[56–58], the heterogeneity of the shell matures faster at higher temperatures, compare for example $\Delta T = 21.8$ and 40.3 K in Fig. 3d. Texture evolves over a period of hours, from smooth, composed of micron sized crystallites, to rougher and more heterogeneous, Fig. 3d and Supplementary Fig. 10. Broad, faceted crystals grow at the expense of small ones, which is typical of Ostwald ripening, Supplementary Fig. 11. Polarization microscopy (useful for ice Ih[34,59]) does not differentiate grains in the cubic sI crystals of methane hydrate. We put traces of the rigidochrome fluorescent dye DASPI in the water for some experiments. DASPI, see Supplementary Fig. 1, does not fit in the clathrate structure, but has higher fluorescence yield when constrained at nm scale towards its flat, fully conjugated configuration, e.g. in micro-porous media[60], viscous liquids[61] or wetting precursor films[51]. Figure 3e compares transmission and fluorescence images of a halo at $\Delta T = 40.3$ K. Consistent with the conclusions of others[37,55], we interpret the coincident bright contours around crystallites in transmission and fluorescence modes as evidence of water diffusing through grain boundaries to feed the lateral expansion and thickening of the halo. Gradual infilling of this porosity is indicated by the fact that the thickness of the halo, $w(t)$, follows empirically $w(t) \propto t^\gamma$, with $\gamma < 1/2$, Fig. 3c, whereas $\gamma = 1/2$ would be expected for diffusion at constant porosity, see Supplementary Note 4. Thickening decelerates faster at high supercooling, consistent with the closure of grain boundaries by crystallization at higher driving force.

Texture governs the tensile response. Figure 3a shows decreases of the elastic modulus, the yield point and the tensile strength, at higher temperatures or smaller $\Delta T$, consistent with the type of failure observed. At high supercooling, failure generally occurs through sharp, often single, regular cracks, running circumferentially round the hydrate sleeve, e.g. Supplementary Video 1, recorded at $\Delta T = 40.3$ K, $t_a = 7$ h. Hydrate grown at higher temperature fails by a more distributed network of small, irregular cracks appearing quasi-simultaneously in several places, e.g. Supplementary Video 1, recorded at $\Delta T = 21.8$ K, $t_a = 7$ h. Figure 3b illustrates the increasing brittle character of hydrate annealed for longer periods, with lower ultimate strength and strain at failure.

## Halo healing after failure.

The cap side of a failed shell snaps to the left in Fig. 1c and a film of water floods over the glass in the break, quickly bearing a secondary halo that spreads over the first. The steps already described for the first halo are repeated. The secondary halo frequently spreads back onto the glass beyond the tip of the first one, cf. Supplementary Fig. 7 and Supplementary Video 1. Recoil of the first halo to the right of the break is very small or not detected, mostly due to accumulated plastic deformation, see e.g. the highlighted tracks of features to the left and right of the breach (the fork) in Supplementary Fig. 3b. Sequential failures with healing may occur during the final warming up at the end of an experiment.

## Discussion

Glass or fused silica micro-capillaries are convenient sample cells for observing and manipulating matter at high pressure and high resolution under the optical microscope. They are of high optical quality, chemically and thermally resistant, strong when handled with care, available in various shapes and sizes,

**Table 1 Tensile properties of the hydrate shell in monotonic loading tests.**

| $\Delta T$ (K) | Replicas | $w(\mu m)$ | $E_{\text{min}}/E_{\text{mean}}/E_{\text{max}}$ (GPa) | $\sigma_t^{\text{min}}/\sigma_t^{\text{mean}}/\sigma_t^{\text{max}}$ (MPa) | $\dot{\sigma}$ ($10^{-2}$ MPas$^{-1}$) | $\bar{\dot{\epsilon}}$ ($10^{-6}$ s$^{-1}$) |
|---|---|---|---|---|---|---|
| 21.8 | 3 | 10.5 | 0.75/0.80/0.86 | 1.9/2.2/2.4 | 0.5 | 9.7 |
| 33.8 | 4 | 9.3 | 1.6/2.2/2.6 | 3.8/4.8/5.7 | 2.4 | 17 |
| 38.3 | 2 | 7.3 | – | 5.9/6.1/6.2 | 3.8 | – |
| 40.3 | 4 | 7.3 | 4.9/5.9/8.1 | 5.2/6.6/7.6 | 5.0 | 24 |

The shell thickness, $w$, the means and ranges of the measured axial elastic modulus, $E$, and of the von Mises equivalent tensile strength, $\sigma_t$, the stress rate, $\dot{\sigma}$, and the average strain rate, $\bar{\dot{\epsilon}}$, are reported for a series of runs at annealing time $t_a = 7$ h, at different supercoolings, $\Delta T = T_{eq} - T_a$, with $T_a$ the temperature of annealing and subsequent tensile test. The equilibrium temperature at the experimental pressure of 15 MPa is $T_{eq} = 16.3\,°C$[52].

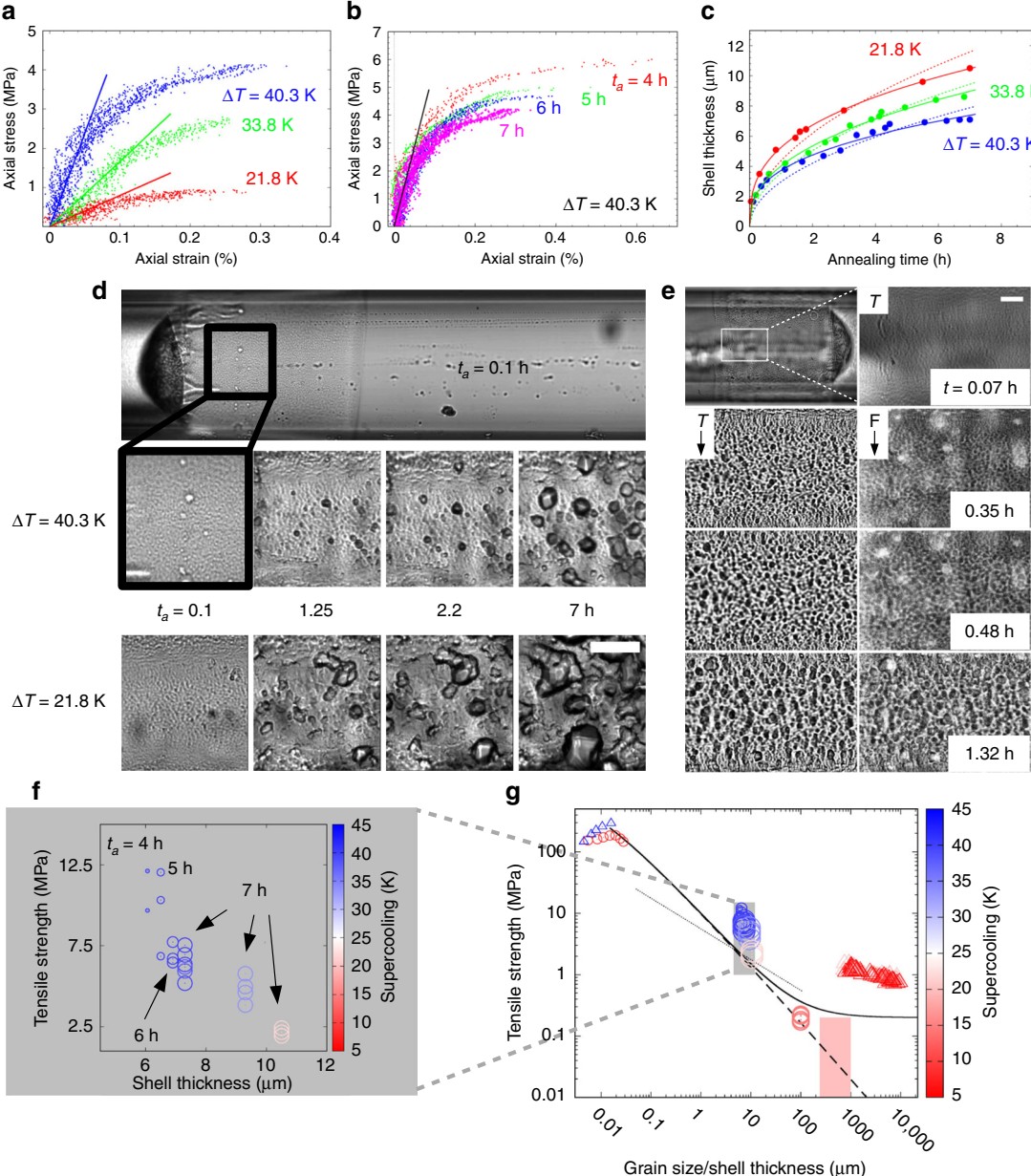

**Fig. 3 Grain-size controls the tensile properties of methane hydrate. a, b** Stress–strain data under monotonic loading to failure, for several points in typical samples. Straight lines in **a** are axial elastic constants $E = 0.8$, 1.6 and 4.9 GPa at supercoolings $\Delta T = 21.8$, 33.8 and 40.3 K; **c** Thickening of the shell at different supercoolings $\Delta T$. Solid lines: $w(t) \propto t^{\gamma}$ with $\gamma = 0.35$, 0.4 and 0.35 at $\Delta T = 21.8$, 33.8 and 40.3 K. Dashed lines: fits to $\gamma = 1/2$; **d** Grain size evolves with both annealing time $t_a$ and supercooling, $\Delta T$. Scale bar: 50 µm; **e** The rigidochrome fluorescent dye DASPI marks grain boundaries in a maturing halo ($\Delta T = 40.3$ K): left transmission images (T), right fluorescence (F). Scale bar 20 µm; **f** Dependence of the von Mises tensile strength on shell thickness, as determined by the supercooling (colour coded pink through ice- to deep blue $\Delta T = 21.8$, 38.3, 33.8 and 40.3 K, and by the annealing time, $t_a$ (symbol size); **g** Situation of the present data (grey box) in a log-log plot of tensile strength vs. grain-size, $g$, (or shell thickness at failure), with respect to experimental (large symbols) and simulated data (small symbols) on methane hydrate (circles refs. [21,24]) and ice (triangles, refs. [25,34]). Supercooling of all data colour coded as in part **f**. Dotted line: slope $\beta = -1/2$ for a standard Hall-Petch relation, Eq. (8); Solid line: size-effect Eq. (9) with $\sigma_{\infty} = 0.2$ MPa, $Y = 11$ GPa[64] and $K = 0.13$; dashed line: predicted extension of the size effect into the range of grain sizes observed in marine sediments (pink box, refs. [33,76]).

and cheap. Their small size and relatively high thermal conductivity are advantageous when thermal cycles are to be applied to the sample.

Such capillaries are used here to grow thin hydrate shells that have features in common with real-world excess-gas situations, including seepage of methane from hydrate-bearing sediments or off-shore installations, particularly hydrate skins encapsulating gas bubbles rising in the water column until rupture by pressure

imbalance. The configuration is also close to the grain- or mineral- coating pore habit of model sediments in presence of an at least local, or pore scale excess of gas[40–42,62].

The hydrate shells show both brittle and ductile tensile features. The brittle character is more pronounced when the hydrate shell is annealed at lower temperatures or for longer times. For example, fracture at high supercooling is mostly localised, typically involving nucleation of a dominant crack, which develops

perpendicular to the stress axis (see picture 6 in Fig. 1c. The total strain at failure is then small and independent of supercooling, ≈0.3%. The decrease of the apparent elastic constant in cyclic loading tests is associated with accumulated damage (fatigue), another characteristic feature of the brittle behaviour of the hydrate, accentuated by prolonged annealing. Nonetheless, strain hardening, usually associated with ductility, is observed under cyclic loading at the deepest supercooling, $\Delta T = 40.3$ K, Fig. 2d.

In relation to earlier work, the data appear to be the first estimation of an elastic modulus of methane hydrate by a traction test at micron scale, albeit unconventional, contactless and requiring a measure of theory and assumptions justified as far as possible here. With these restrictions, the present method gives access to the non-linear regime beyond the small displacements probed by acoustic or Brillouin scattering experiments. The elastic modulus of polycrystalline hydrate increases with super-cooling, from 0.8 to 5.9 GPa, between $\Delta T = 21.8$ and $\Delta T = 40.3$ K (Table 1). As expected, it is smaller than the range 11–14 GPa obtained for single crystals by Brillouin scattering, albeit at lower supercooling[63,64]. Practical difficulties of mechanical tests on gas hydrates encouraged studies with molecular modelling. The elastic modulus of monocrystals ranges from ≈11 GPa in ab initio calculations[65] to between 9.7 and 7.7 GPa, in the temperature range 200–283 K in classical molecular dynamics simulations[24]. MD simulations on polycrystalline methane hydrate at ≈10 nm scale[24] yield Young's moduli ≈6.3 GPa, below those of the model mono-crystals in that study, as expected. Experimental studies of the strength of methane hydrate mostly report data under compression, ranging from ≈2–10 MPa at high strain rate[66,67] to ≈50–100 MPa at low rates[31,32]. To our knowledge, the only previous measured tensile strength is ≈0.2 MPa, for a mm-sized sample grown between calcite plates[21]. Here, we find equivalent von Mises tensile strength between ≈2 MPa at $\Delta T = 21.8$ K and ≈7 MPa at $\Delta T = 40.3$ K, at low strain rates similar to the compressive tests of refs. [24,31,32] reports compressive and tensile strengths (strictly maximum stresses) of order 150–200 MPa in MD simulations of polycrystalline nm-scale models, again lower than for a single crystal in the same study (≈800 MPa).

Contrary to methane hydrate bearing sediments, experimental data on the strength of the hydrate itself are thus sparse and restricted almost entirely to compressive tests. There appears to be a large discrepancy between simulations and experiment. But grain size strongly affects the mechanical properties of poly-crystalline materials by the Hall-Petch effect: the smaller the grains the stronger the material, except for the inverse effect at nm scale. The classical connection between grain size, $g$ and tensile strength, $\sigma_t$, or more often the yield stress, $\sigma_y$, is the Hall-Petch relation[27,28]. In relation to ice, see for example ref. [68]. Here, we cast the relation in the form

$$\sigma_t(g) = \sigma_\infty \left(1 + (g_0/g)^\beta\right), \qquad (8)$$

where $\sigma_\infty$ is the strength at large grain size and $g_0$, is an inflection point dependent on the material and the temperature. Exponent $\beta = 1/2$ was justified by consideration of dislocation pile-up at grain boundaries[69] and applied widely down the years to many data sets and materials. But because of the difficulty of testing a power law relation over limited data ranges for given systems, its statistical and theoretical significances are increasingly questioned, e.g. exponents $\beta$ ranging at least between 0.2 and 1 may be equally valid statistically and parameters for closely related materials may differ widely[29]. The size effect, dating to the work of Bragg[70], relates inversely the space available

(grain size) to the stress required for production of dislocations, leading to[29,30,71–73],

$$\sigma_y(s)/Y = \sigma_\infty/Y + K\frac{\ln(s/a_0)}{s/a_0}, \qquad (9)$$

where $\sigma_y$ is the yield stress and $Y$ is Young's modulus (so $\sigma_y/Y$ is an elastic strain), $\sigma_\infty$ is the yield stress for very large grains, $a_0$ is the unit cell parameter (standing in for the Burgers vector) and $K$ is a dimensionless constant expected to be around unity. Variable $s$ is an effective size, the harmonic mean of the grain size and the device size[74], here the halo thickness, $w$, $s^{-1} = g^{-1} + w^{-1}$. The heterogeneity of grain size increases with time in our samples, but in the quasi-two dimensional halo, it is the size of the smaller grains which determines their area of contact with the larger ones, see Fig. 3e and Supplementary Fig. 10f, making halo thickness a reasonable proxy.

How does the Hall-Petch effect influence gas hydrates? Figure 3 of ref. [24] plots strength vs. grain size for simulated polycrystalline methane hydrate at scale ≈10 nm, and in the absence of experimental data, those for polycrystalline ice at scale 1–10 mm[34]. The plot suggests the Hall-Petch effect above ≈10 nm, but ice is a poor proxy for methane hydrate, with e.g. more brittle fracture found here for smaller grained samples vs. more ductile behaviour in ice[34]. Varying methods of sample preparation may be responsible for part of the differences, but there are differences between the mechanical properties of ice and methane hydrate, cf. refs. [31,32,59,75] and the discussion in ref. [25]. Although there is a data gap of orders of magnitude between theory and experiment in the plot, the fair agreement of the simulated elastic constants of monocrystals in ref. [24] with the experimental data[63,64] and values simulated with other methods[65], suggests that the models may be used for the present as proxies for the tensile strength of polycrystalline methane hydrate at nm scale. Ref. [21] does not report the grain size, but we estimate at most ≈100 μm, based on inspection of its Fig. 2d–f and consistent with the values reported for synthetic methane hydrate, which has smaller grains than natural samples[33,76]. Considering the range of sizes to be covered, we therefore put together in the log-log plot of Fig. 3g the simulations of ref. [24], the data of ref. [21] and the present data, where we should as far as possible compare systems with similar supercooling.

The plot confirms that grain-textured ice is not an acceptable analogue for polycrystalline methane hydrate. Further, the standard Hall-Petch relation, Eq. (8) with $\beta = -1/2$ is incompatible with the data for methane hydrate over so wide a range. The size effect, Eq. (9) with $\sigma_\infty = 0.2$ MPa better connects the present data at low supercooling (red/pink points) and the simulations of ref. [24]. But either the limiting tensile strength is less than the ≈0.2 MPa in ref. [21] (tailing off of the solid line), or one must assume an untenable grain size, ≈1 mm. Assuming a smaller ultimate stress (here in the absence of data, vanishing $\sigma_\infty$, dashed line), accounts better for all the available data. Whatever the final value of $\sigma_\infty$, such agreement is astonishing, considering the very different numerical and experimental methods and sample configurations used for determining this property. Extension of the size effect to the larger grains found in marine sediments (pink box in Fig. 3g)[33,76] suggests the contribution of the cohesion of methane hydrate in marine and geological sediments could be up to an order of magnitude lower than presently supposed. Measurement of the tensile strength of samples with larger grain sizes is therefore important. Gas hydrate in geological or marine settings is a complicated material, subject to numerous influences, such as creep (albeit less susceptible than ice[32,34]), ongoing crystallization and dissociation processes in presence of varying

gas concentration in marine currents, burial under sediments, action of micro-organisms. But, considering the present state of the data, geological time scales and the greater thermodynamic stability of larger grains over smaller ones, despite kinetic barriers, the cohesion of methane hydrate in such sediments appears overestimated and piecemeal destined to diminish.

## Methods

Full details of materials and methods are provided in Supplementary Notes 1–3.

**Materials**. We use 10 cm long fused silica capillaries with internal and external diameters 200 and 330 μm (Vitrotubes, CMScientific), with deionized water (Purelab classic system, electrical resistivity 18 MΩcm$^{-1}$) and 99.9995% grade methane (Linde).

**Methods**. Capillaries are glued into 1/16″ steel tube, set in a three-way valve (TopIndustrie), with an ISCO DM65 syringe pump to control the gas pressure. For transmission images, the capillary is observed in a cooling and heating stage (Linkam Cap500 with Linksys software) on an Olympus BX50 upright microscope stand, with a ×10 extra-long working distance objective (Olympus) and a Ueye UI 3360 camera run mostly at 1 frame per second. Fluorescence and higher resolution transmission images are acquired with a home-made stage and a custom thermostat (Étincelage) on an inverted stand (Nikon Ti-eclipse) with a ×20 extra long working distance objective with a correction ring, LED illumination (Thorlabs), appropriate filters for DASPI (Semrock) and an ORCA 4.0 camera (Hamamatsu). Image processing with Fiji[77] and data analysis with gnuplot[78] are used to extract the displacement of crystal grains from the videos and determine the strain field in the hydrate shell.

## Data availability

On reasonable request, original data reported in this paper are available from the authors. Original data for Supplementary Videos 1 and 2 are available at: https://doi.org/10.17632/ynm22j66cx.2.

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

## Acknowledgements

We acknowledge helpful discussions with A. M.Tang and P. Dangla, Laboratoire Navier at École des Ponts Paristech and D. J. Dunstan, School of Physics, Queen Mary University of London, and help setting up the experiments from A. Touil and J. Diaz, LFC-R, Université de Pau. This work was supported by grants from l'Agence Nationale pour la Recherche (HYDRE 15-CE-06-000) and la Communauté d'agglomération Pau Béarn Pyrénées (Laboratoire en capillaire). Video data storage and computer time for this study were provided by the MCIA (Mésocentre de Calcul Intensif Aquitain) of Université de Bordeaux and Université de Pau et des Pays de l'Adour.

## Author contributions

D.A. and D.B. conceived the study. D.A. and R.B. performed the experiments. D.A., D.B., J.-M.P. and R.B. all contributed to data analysis and writing the paper.

## Competing interests

The authors declare no competing interests.
