## [Peer Review File · Nature Communications]

Reviewers' comments:

Reviewer #1 (Remarks to the Author):

In this paper, the authors reported the growth of thin films of the hydrate at 15MPa in thin glass capillaries, and they showed that the thermal-induced stress results in the failure of their prepared hydrate samples (hydrate shell). However, there are several important issues the authors needed to address further before publication in Nature Communications.

(a) My first important concern: How to evaluate the feasibility and availability of the experimental method in obtaining the failure properties of hydrate shell in this manuscript? In other words, the authors should show the converged results. How much will the thermal-induced lattice expansion of hydrate shell and the thin glass capillary affect the present results? It should be noted that the small expansion of the thin glass may produce large internal stress on the hydrate shell. This internal stress may bring about important effect on the structure of the hydrate shell. The authors should give the change in the radius of the glass capillary as the temperature increases. How much will the structure of the hydrate shell be affected by the temperature change?

(b) My second important concern: As shown in the paper (Page 19), the grain size is a key parameter which will influence the mechanical properties of polycrystalline materials. Besides grain size, grain morphology texture, hydrate sample purity, and hydrate phase structure (sI, sII, and sH) can also play an important role in the mechanical strength of polycrystalline hydrates. The information is unclear. As shown in the paper and supplementary videos, the grain size of their prepared samples fluctuates greatly. How to determine these impacts on the present results?

(c) My third important concern: In their experiments, the contacting force between their grown hydrate shell and the thin glass capillary seems to be larger than those of hydrate shell itself. As pointed in the paper in Page 5, a thin water layer is found between micron-thin hydrate shell and the sediment particles, how to understand the contacting interface structure between hydrate shell and the thin glass capillary in this paper?

There are also other issues as bellows:

(1) The writing of this paper is not good at all. There are a lot of mistakes in English writing. And the presentation of this paper is also not good, many parts of this paper need to be rewritten.

(2) In Page 1, "... and tensile strength as functions of the parameters controlling grain size...", the controlled parameter may be the thickness of hydrate shell, not grain size. The thickness of hydrate shell and the grain size are two different parameters. This is absolutely misleading. Thus, using the shell thickness as a proxy for grain size is not correct.

(3) In Page 6, "...by creating a greater pressure on its gas than on its water side...", it makes readers very confused because this description is contradictory with the content in Page 1 (...A novel method of thermally induced stress generation enables contactless characterization of the tensile).

Reviewer #2 (Remarks to the Author):

The manuscript presents an elegant experimental approach to study a complex and difficult material. The following comments are triggered by observations reported in the manuscript (some of them unprecedented – e.g., Figure 2). The emphasis is on potential pitfalls and alternative interpretations rather than the many lessons learned.

KEY QUESTIONS/ISSUES

- Neat experimental study leading to surprising measurements. The limited discussion in the text hides inherent experimental complexities and associated uncertainties in indirect measurements

(some discussed in supplementary material). In this context, I wonder about the distinction between “what we think we measure, what we want to measure and what we measure”. The discussion section should address experimental challenges and uncertainty propagation before any in-depth physical interpretation (e.g., Hall-Petch effect). This review attempts to identify some potential issues.

- All tests are run at high degree of supercooling, i.e., below water freezing. Therefore, the coexistence of ice-and-hydrate and experimental implications must be carefully addressed.
- What is the meaning of tensile strength in a material that is pressure sensitive? In other words, what is the difference between fluid pressure and solid stress at the atomic scale and their implications on hydrate stability? Could the tensile failure of hydrate mass result from local hydrate dissociation during tensile loading due to the effective decrease in internal stress? Such a possible interpretation was raised in Jung and Santamarina (2011). If this were the case, the high supercooling conditions used in this study could explain -at least in part- the higher tensile strengths reported by the authors.

OTHER COMMENTS

Note: references to our own work are added to several comments, NOT to promote our work (in fact, the authors have already recognized our efforts generously), but to provide clarity and support to my comments (furthermore, our papers cite articles of potential interest to the authors).

- Page 1: requires careful editing
- Page 2:
 - o Strength: hydrate effects dilation in short-term tests (the effect depends on pore habit – see various DEM studies, such as Brugada et al 2010 – Granular matter; Jung et al., 2012 JGR).
 - o Hydrate creep may be more relevant than strength to long-term stability (Refer to page 21).
- Page 5&6
 - o Why run the system at such a low temperature $T=-23^{\circ}\text{C}$? Clearly, the induction time will be shorter, but it drives hydrate formation unrealistically fast after nucleation (e.g., consider crystal growth, the local effects of exothermic heat, diffusion from the initially saturated water phase, etc).
 - o Ice: the authors do not report ice formation. As freezing point depression can be disregarded for this pore size, I wonder whether I am missing something critical. Line 5 on pg 6 does not clarify the matter. Induction times may be long under quiescent conditions, but active hydrate formation should be enough to trigger ice formation - or, in fact, it may be the other way around (Dai et al. 2014, Fluid Phase Equilibria 378, 107–112). Any evidence of ice content? Raman data?
 - o Annealing by “slightly raising the temperature”: how much? As you started at $T=-23^{\circ}\text{C}$, I assume you are still much lower than $T=0^{\circ}\text{C}$? If so, why not anneal at $T>0^{\circ}\text{C}$ to make sure there is no ice? (I can see implications on volume contraction, but the alternative is concerning).
 - o Could the authors discuss hydrate wettability on borosilicate (data from the literature may help). Consequences of rapid transformation? Evidence of tip adherence? Implications on results.
 - o Did the authors observe moisture migration and condensation ahead of the meniscus, particularly during the first stage at 20°C (it will probably diminish at -23°C - see Sun et al., 2018. Time-dependent pore filling. Water Resources Research)? This will have important implications on the halo formation and does not require water film effects and transport. The movie may support this mechanism: as soon as the seal breaks during tension, vapor transport/condensation could sustain hydrate formation ahead of the initial halo.
 - o Young hydrate is porous and permeable (also page 9 - Evidence and references in Liang and Santamarina JGR 2018; also observed in other crystallizations, e.g., Dai et al 2016 – Acta Geotechnica)
 - o End of page 6: Ostwald ripening (ok: listed on page 13)

- Page 7

- o “The most obvious is to increase the pressure of the gas”: why would this work for hydrate resting on stiff water? expansion of the tube? (note: both longitudinal and transverse)?
- o Increase temperature: water contraction and simultaneous glass expansion/extension (ok: addressed on page 8)
- o Failure appears to happen away from the cap. Why? Tension decreases away from the cap due to hydrate-wall interaction, doesn't it?

- Page 8

- o The measurement of displacements (described in S1): it appears to be a featured-centered digital image correlation – correct?. Please mention in the manuscript. Critical to all analyses

- Page 9

- o “Crystal growth is not significantly influenced by the applied stress”: explain

- Page 10

- o Supercooling temperatures: once again, address issues related to the apparent absence of ice.

- Page 15

- o Water flow without freezing or forming hydrate: it implies that the rate of transport is faster than rate of transformation - Damkohler (also observed in Jung and Santamarina, 2012, J. Crystal Growth 345 61–68). Related discussion?

- Page 16

- o Mostly summary of previous parts (clearer and better written). Avoid duplication.

- Page 17

- o Indeed, hydrate growth reported in this study is particularly relevant to excess-gas / water-limited conditions (e.g., at the base of the BSR, next to faults and gas-release features such as pipes)

- Page 21

- o Considering geological time scales: hydrate creep will be critical on the long-term behavior of hydrate bearing systems

EDITORIAL

- Abstract and some figure captions: require careful editing.
- The text could be tighter (it could afford a 40% shortening without loss of information)

Contactless measurement of the tensile properties of methane hydrate at pore scale

by D. Atig *et al.*

Reply to reviewers' comments

Replies are indented for clarity

Page numbers in italics at the end of each reply refer to the revised MS

We thank both reviewers for thoughtful and constructive comments and apologize for the delay in the reply (which we would have wished shorter for the sake of their time), due to factors unrelated to the paper.

Reviewer #1 (Remarks to the Author):

In this paper, the authors reported the growth of thin films of the hydrate at 15MPa in thin glass capillaries, and they showed that the thermal-induced stress results in the failure of their prepared hydrate samples (hydrate shell). However, there are several important issues the authors needed to address further before publication in Nature Communications.

(a) My first important concern: How to evaluate the feasibility and availability of the experimental method in obtaining the failure properties of hydrate shell in this manuscript? In other words, the authors should show the converged results. How much will the thermal-induced lattice expansion of hydrate shell and the thin glass capillary affect the present results? It should be noted that the small expansion of the thin glass may produce large internal stress on the hydrate shell. This internal stress may bring about important effect on the structure of the hydrate shell. The authors should give the change in the radius of the glass capillary as the temperature increases. How much will the structure of the hydrate shell be affected by the temperature change?

This question raises three points, all addressed at appropriate places in the revised MS:

(i) Does differential thermal strain contribute to the failure of the hydrate? We make it clearer that the hydrate shell binds to the glass only at its tip. Elsewhere, it stands free of the glass, separated from it by the water film, typically 10 μ m thick. Differential expansion is therefore not the cause of failure of the hydrate, because failure is always observed in the free standing part of the shell, not in the area adhering to the glass. **(Pp. 4-10 contain a re-written, re-ordered, augmented description of the formation of the hydrate shell.)**

This said, the change of radius of the glass capillary is actually very small. Considering typical temperature excursions to induce failure, $\delta T=2$ K, the change in size of the bore of the capillary (radius R) is $\delta R/R = \alpha_G \delta T$, where $\alpha_G \approx 6 \times 10^{-6} \text{ K}^{-1}$ is the coefficient of thermal expansion of the glass. Thus $\delta R/R < \sim 10^{-5}$, or $\delta R \approx 1$ nm for $R=100\mu\text{m}$. Even at the point of adhesion, the possible *differential* strain, $\varepsilon_T = |\alpha_H - \alpha_G| \delta T < 10^{-4}$, is an order of magnitude smaller than the strain at failure, $> \sim 10^{-3}$ (using the thermal expansion of methane hydrate, $\alpha_H = 70 \times 10^{-6} \text{ K}^{-1}$ [Sloan, 2008]). **(Addressed middle paragraph p. 10)**

(ii) Does thermal expansion contribute to the observed deformation? No, it is an order of magnitude smaller than the observed strain, see above. **(Also p. 10)**

(iii) Finally, the experimental conditions are well within the stability zone of CH₄ hydrate sI, so the small temperature change during the tensile test is not expected to change the crystal structure of the hydrate (we are aware of only one report of kinetically trapped sII methane hydrate, seen under milder conditions, and that converted to sI, the thermodynamically more stable form[Schicks2004]). **(Also p. 10)**

(b) My second important concern: As shown in the paper (Page 19), the grain size is a key parameter which will influence the mechanical properties of polycrystalline materials. Besides grain size, grain morphology texture, hydrate sample purity, and hydrate phase structure (sI, sII, and sH) can also play an important role in the mechanical strength of polycrystalline hydrates. The information is unclear. As shown in the paper and supplementary videos, the grain size of their prepared samples fluctuates greatly. How to determine these impacts on the present results?

Under the present experimental conditions (15MPa, -23.5 to -5°C), sI is the stable crystal structure. Under milder conditions (closer to dissociation) form sII may be kinetically trapped, but converts to sI[Schicks2004]. **(Text amended p. 10.)**

Under any given conditions, the samples comprise relatively few, but large, faceted, plates, domed towards the gas side of the halo, that emerge from areas of smaller crystals. The effective cross-section of the halo is thus that of the smaller crystals (contacting the bases of the thicker and broader crystals). Breaks occur in the areas of smaller crystals, not across the larger monocrystals. **(Sentence starting “Polyhedral crystallites...” end of l. 10, p. 5, to end of paragraph. See also the reply to other issue no. 2.)**

(c) My third important concern: In their experiments, the contacting force between their grown hydrate shell and the thin glass capillary seems to be larger than those of hydrate shell itself. As pointed in the paper in Page 5, a thin water layer is found between micron-thin hydrate shell and the sediment particles, how to understand the contacting interface structure between hydrate shell and the thin glass capillary in this paper?

As made clearer in the revised MS, the halo is like an inner sleeve in the capillary, contacting the glass at the halo-tip (on the right in all our figures), whereas failure occurs closer to the cap (former meniscus, on the left), where the halo stands free of the glass, separated from it by a layer of water. We are therefore measuring the cohesion of the sleeve, not its adhesion to the glass, similar to the observations of Jung & Santamarina, for methane hydrate grown between calcite plates [*Geochem. Geophys. Geosys.* 2011]. **(Pp. 4-10 contain a re-written, re-ordered, augmented description of the formation of the hydrate shell.)**

There are also other issues as bellows:

(1) The writing of this paper is not good at all. There are a lot of mistakes in English writing. And the presentation of this paper is also not good, many parts of this paper need to be rewritten.

Whole paper overhauled.

(2) In Page 1, "... and tensile strength as functions of the parameters controlling grain size...", the controlled parameter may be the thickness of hydrate shell, not grain size. The thickness of hydrate shell and the grain size are two different parameters. This is absolutely misleading. Thus, using the shell thickness as a proxy for grain size is not correct.

To continue the discussion started over major concern (b) above, since submitting the MS, our attention has been drawn to a recent review (D. J. Dunstan, *J. Mater. Res.* **32(21)** (2017) 4041-4053). That paper and [Dunstan, *Phys. Rev. Lett.* 2009], both added to the references, discuss how the size effect on materials' strength is determined by the smaller of the grain size, g , and the device (or sample) size, d (here the thickness of the halo, w), via an effective harmonic scale s , such that $1/s = 1/g + 1/d$. In the quasi-2d halo (cf. pp. 4-10), the contact between the large and the small grains is determined by the size of the latter, equivalent to the shell thickness, w , since the shell comprises one or two layers of small crystallites out of which the large ones emerge (figure 10 of the SI). **(Paragraph continuing under eq. (10), p. 18 provides a short explanation.)**

(3) In Page 6, "...by creating a greater pressure on its gas than on its water side...", it makes readers very confused because this description is contradictory with the content in Page 1 (...A novel method of thermally induced stress generation enables contactless characterization of the tensile).

Amended for clarity to "We put the shell under tensile stress by contriving a lower pressure on the water than on the gas side, by slightly increasing the temperature at constant gas pressure, say a temperature excursion $\delta T(t)$ at time t with respect to T_a . Since T_a is below the inversion temperature..." **(Para. starting last line p. 6. Also see below the reply to reviewer 2's other comment on p. 7 of the original MS)**

Reviewer #2 (Remarks to the Author):

The manuscript presents an elegant experimental approach to study a complex and difficult material. The following comments are triggered by observations reported in the manuscript (some of them unprecedented – e.g., Figure 2). The emphasis is on potential pitfalls and alternative interpretations rather than the many lessons learned.

KEY QUESTIONS/ISSUES

- Neat experimental study leading to surprising measurements. The limited discussion in the text hides inherent experimental complexities and associated uncertainties in indirect measurements (some discussed in supplementary material). In this context, I wonder about the distinction between "what we think we measure, what we want to measure and what we measure". The discussion section should address experimental challenges and uncertainty propagation before any in-depth

physical interpretation (e.g., Hall-Petch effect). This review attempts to identify some potential issues.

The spread of values of the tensile properties in table 1 provides an indication of the uncertainties of “what we measure”. For the relatively small numbers of replicas (remember these are difficult and long experiments), we prefer to quote the mean and range rather than the standard deviation. The ranges relative to the mean values suggest we really are measuring something, albeit with uncertainties probably larger than in more classical mechanical tests. (Even there, discussion of errors may be eluded.) But we feel the usefulness of the data is more their contribution to the very wide gap in our present knowledge of the mechanical properties of methane hydrate. **(Table 1 p. 12)**

While on the subject of “what we measure”: Considering the reviewer’s comments, we thought it helpful to specify the complete estimated stress state of the halo, and to facilitate comparison with the literature by reporting the tensile strength as equivalent von Mises tensile stress at failure. We also point out that what we plot in the graphs of elastic data is just an axial modulus, since the thinning of the stretched halo is below detection so neither the Poisson ratio nor the Young’s modulus may be derived. **(Eqns. 4-10, pp. 8-9)**

Further questions on “what we want to measure“ are addressed in the point by point discussion below.

•All tests are run at high degree of supercooling, i.e., below water freezing. Therefore, the coexistence of ice-and-hydrate and experimental implications must be carefully addressed.

First, regarding the samples for the data presented: Despite close scrutiny during the final warming up to room temperature at the end of an experiment, no particular change in the appearance of the samples is observed at the expected melting point of ice. Second, very occasionally, ice *does* nucleate at low temperature ($\approx -20^\circ\text{C}$), *before* nucleation of the hydrate. But the presence of ice is as spectacular as it is unmistakable. The whole water column explosively transforms ($\approx 1\text{s}$) to opalescent ice encapsulating bubbles of methane. This solid indeed melts at the melting point of ice. The revised presentation of sample preparation makes this clearer and mentions that such runs were abandoned. **(See middle paragraph p. 9)**

•What is the meaning of tensile strength in a material that is pressure sensitive? In other words, what is the difference between fluid pressure and solid stress at the atomic scale and their implications on hydrate stability? Could the tensile failure of hydrate mass result from local hydrate dissociation during tensile loading due to the effective decrease in internal stress? Such a possible interpretation was raised in Jung and Santamarina (2011). If this were the case, the high supercooling conditions used in this study could explain -at least in part- the higher tensile strengths reported by the authors

Indeed, but there are two differences between the sample configuration in [Jung & Santamarina 2011] and the present samples. First, the former, yoyo-disk shaped samples were examined closer to the phase boundary, and second they were much larger than the thin hydrate sleeve used here. There, traction on the flat disk faces could indeed cause the internal stress in the middle of the disk to drop below the

phase boundary of the hydrate, thus could trigger dissociation. Here, the thin hydrate halo is subjected to (i) “what we think we measure”, viz. the (uniaxial) stress exerted by traction on the cap at the former meniscus, and (ii), to internal transverse and circumferential stresses, caused by the difference in the hydrostatic pressures of the gas on the inside, p_g (constant $p_g=15\text{MPa}$), and the water on the outside, $p_w \approx p_g - (\alpha_w/\beta_w)\delta T$. Because the (projected) area of the cap is bigger than the section of the halo, we gain a mechanical advantage needing only a modest pressure difference to break the halo, $p_g - p_w < 1\text{MPa}$. The axial stress is also small ($\approx 3\text{MPa}$) so overall the samples remain well within the domain of stability. **(Paragraph below eqn. 6, p. 8 and figure 8 added in the revised SI, comparing all stress components to the phase boundary)**

OTHER COMMENTS

Note: references to our own work are added to several comments, NOT to promote our work (in fact, the authors have already recognized our efforts generously), but to provide clarity and support to my comments (furthermore, our papers cite articles of potential interest to the authors).

- Page 1: requires careful editing

Abstract revised and shortened. **(p. 1)**

- Page 2:

o Strength: hydrate effects dilation in short-term tests (the effect depends on pore habit – see various DEM studies, such as Brugada et al 2010 – Granular matter; Jung et al., 2012 JGR).

Included in the references, as making it all the more surprising that our knowledge of the hydrate itself is still very incomplete. **(cited l. 3, p. 3)**

o Hydrate creep may be more relevant than strength to long-term stability (Refer to page 21).

See below the reply to the reviewer’s comment on p. 21 of the original MS.

- Page 5&6

o Why run the system at such a low temperature $T=-23\text{ }^\circ\text{C}$? Clearly, the induction time will be shorter, but it drives hydrate formation unrealistically fast after nucleation (e.g., consider crystal growth, the local effects of exothermic heat, diffusion from the initially saturated water phase, etc).

The new MS should be clearer on this point: The smallness of the samples is the reason a high degree of supercooling is necessary to initiate formation of the hydrate. We quench until nucleation on the meniscus, in practice usually around -23°C . So the very first crystallites in all samples are similar. But we quickly warm the sample to the annealing temperature, T_a , where it is left for a period of hours (the highest was $T_a = -5^\circ\text{C}$). Grain size differentiates as a function of annealing temperature, but

all samples appear to be converging at different *rates* towards similar textures (due to Ostwald ripening for example). (**See paragraph starting p. 4 & continuing top of p. 5.**)

o Ice: the authors do not report ice formation. As freezing point depression can be disregarded for this pore size, I wonder whether I am missing something critical. Line 5 on pg 6 does not clarify the matter. Induction times may be long under quiescent conditions, but active hydrate formation should be enough to trigger ice formation - or, in fact, it may be the other way around (Dai et al. 2014, Fluid Phase Equilibria 378, 107–112). Any evidence of ice content?

No evidence. See also the reply to "KEY ISSUES". The revised MS should dispel any ambiguity. Most of the time, CH₄ (and incidentally, in other experiments, CO₂) hydrate nucleates before ice. When ice appears first (rare), we abandon the run. (**See middle paragraph, p. 9.**)

o Annealing by “slightly raising the temperature”: how much? As you started at T=-23C, I assume you are still much lower than T=0C? If so, why not anneal at T>0C to make sure there is no ice? (I can see implications on volume contraction, but the alternative is concerning).

This source of misunderstanding should be avoided by the wording of the revised text. (**See end of 1st paragraph p. 5.**)

o Could the authors discuss hydrate wettability on borosilicate (data from the literature may help). Consequences of rapid transformation? Evidence of tip adherence? Implications on results.

Any slip of the hydrate tip was below detection (<≈ 1μm from our image processing). Non-adherence would cause the whole hydrate shell to slide left on lowering the pressure in the water (not observed).

o Did the authors observe moisture migration and condensation ahead of the meniscus, particularly during the first stage at 20C (it will probably diminish at -23C – see Sun et al., 2018. Time-dependent pore filling. Water Resources Research)? This will have important implications on the halo formation and does not require water film effects and transport. The movie may support this mechanism: as soon as the seal breaks during tension, vapor transport/condensation could sustain hydrate formation ahead of the initial halo.

This is an interesting and a many faceted topic. The short answer is that scrutiny of the videos shows that when the first halo breaks, the shell on the water side snaps to the left (restoring the initial pressure in the water compartment). A (thin!) wave of clearly *liquid* water escapes, frequently seen as a shadow passing along the capillary, on which the second halo quickly (in seconds) may be detected re-covering the breach. (**Commentary of video 1, p. 22 of the SI**)

Breath figures on the glass: Contrary to our earlier work on cyclopentane hydrate halos [Martínez de Baños 2016], no condensation ahead of the meniscus was detected (mentioned in the revised MS). Two reasons may be put forward. Here, the glass exposed to the guest phase is only mildly hydrophilic (as received), with a contact angle ≈15°, compared to <1° for the cold plasma-treated glass in the former study.

Secondly, the amount of water in gaseous methane at the start of the experiment is much lower than that in liquid cyclopentane, hence also the capacity to form breath figures on quenching the system. (**Discussion starting top p. 10**)

A further indication of the small contribution of any breath figure is that the present halos are ridged *perpendicular* to the direction of spreading, whereas individual droplets in the clearly visible breath figure in [Martínez de Baños 2016] produced ridges *parallel* to the direction of growth. (**p. 10 again**)

o Young hydrate is porous and permeable (also page 9 - Evidence and references in Liang and Santamarina JGR 2018; also observed in other crystallizations, e.g., Dai et al 2016 – Acta Geotechnica)

References added in the description of the formation of the hydrate shell (**Line 10, p. 5**).

o End of page 6: Ostwald ripening (ok: listed on page 13)

(and we endeavour to separate observation and interpretation)

•Page 7

o “The most obvious is to increase the pressure of the gas”: why would this work for hydrate resting on stiff water? expansion of the tube? (note: both longitudinal and transverse)?

Indeed, it doesn't work! At least not reproducibly. We dropped this in the revised MS, as unnecessarily distracting the reader.

o Increase temperature: water contraction and simultaneous glass expansion/extension (ok: addressed on page 8)

(and discussed in detail in section 1.5 of the SI, on measuring the compressibility and thermal expansion *in situ*.) (**p. 8 of the SI**)

o Failure appears to happen away from the cap. Why? Tension decreases away from the cap due to hydrate-wall interaction, doesn't it?

Except in rare cases (*e.g.* impatient experimenters jogging a juvenile hydrate shell by moving the capillary too sharply on the microscope!), failure always occurs away

from the cap, where the hydrate does not touch the wall. The cap, as a spherical surface, is stronger than an approximately cylindrical halo of the same material and thickness. ***(Should be clearer from the revised description of the growth and testing of the halo, pp. 4-10)***

- Page 8

- o The measurement of displacements (described in S1): it appears to be a featured-centered digital image correlation – correct? Please mention in the manuscript. Critical to all analyses

Two reasons motivated a different, home-made approach. First, only a narrow longitudinal strip of the halo can be in sharp focus (aberration due to the round capillary). Second, the data representation as false colour maps of intensity along a line of pixels as a function of time makes failure and other events (plasticity) stand out very clearly. So no, the present method is sadly less sophisticated than digital image correlation, but we believe adequate. We devised an algorithm to identify “sufficiently sharp” intensity maxima along a line of in focus pixels in the first image, parallel to the direction of growth/traction. In practice these features are mostly grain boundaries. We fit a parabola to each maximum and use the peak of the parabola to define the position of the feature. Parameters of the fit are used to start the fit in the next image and so on. ***(Reinforced on p. 6 of the revised SI.)***

- Page 9

- o “Crystal growth is not significantly influenced by the applied stress”: explain

Amended to: ...neglecting any influence of stress (micro-cracks) on the rate of crystal growth during tensile tests, because they are short compared to the annealing time... ***(Just below eqn. 8 , p. 10)***

- Page 10

- o Supercooling temperatures: once again, address issues related to the apparent absence of ice.

Done above.

- Page 15

- o Water flow without freezing or forming hydrate: it implies that the rate of transport is faster than rate of transformation - Damkohler (also observed in Jung and Santamarina, 2012, J. Crystal Growth 345 61–68).

The concentration of dissolved gas in the water behind the cap is depleted by the formation of methane hydrate needles. Consistent with this, (i) this water flows

between the halo and the glass without forming hydrate; (ii) the halo thickens predominantly on the gas side, fed by water seeping through grain boundaries *cf.* [Davies2010, Liu2019] and the present data with DASPI. But at the tip, the water forms a meniscus stretched between the glass and the growing edge. Lateral growth of the halo is fed by gas dissolving into the meniscus, with a local concentration $c \approx c_{gw}$, the equilibrium concentration of methane in water, and then diffusing down the concentration gradient to the halo tip, where $c \approx c_{hw} < c_{gw}$ the equilibrium concentration of methane in water in presence of the hydrate, *cf.* [Mochizuki2017].
(Last paragraph, p. 9)

- Page 16

- o Mostly summary of previous parts (clearer and better written).

The revised MS avoids duplication

- Page 17

- o Indeed, hydrate growth reported in this study is particularly relevant to excess-gas / water-limited conditions (e.g., at the base of the BSR, next to faults and gas-release features such as pipes)

Added to the perspective in the Discussion (**1st paragraph, p. 16.**)

- Page 21

- o Considering geological time scales: hydrate creep will be critical on the long-term behavior of hydrate bearing systems

The closing of the original MS was voluntarily provocative on this point. We agree that the perspective is improved by including this remark, and incidentally mention that methane hydrate is less susceptible to creep than ice [Durham2003, Schulson19984]. In marine or geological settings hydrate may further be modified by material fluxes-- more or less gas-saturated water for example, influence of micro-organisms,... This said, we feel that creep does not exclude recrystallization.
(Final paragraph, starting bottom p. 20)

- Abstract and some figure captions: require careful editing.

Done

- The text could be tighter (it could afford a 40% shortening without loss of information)

The referees' questions prompted considerable additions, that we agree were warranted. The revised text *per se* is nonetheless about 15% (three pages) shorter than before.

REVIEWERS' COMMENTS:

Reviewer #3 (Remarks to the Author):

The authors have appropriately answers to all may major and minor questions. Since my general opinion about the paper was positive and leaning towards publication, now I could firmly recommend to accept th epaper as it is.

Weaker than thought? Contactless tensile probing of polycrystalline methane hydrate at sediment pore scale (*title revised at editorial request*)

by D. Atig *et al.*

Reply to round 2 of the reviewers' comments

Replies are indented for clarity

Page numbers in italics at the end of each reply refer to the 2nd revision of the MS

Reviewer #1 (Remarks to the Author):

The authors addressed all my concerns. Published as is.

Reviewer #2 (Remarks to the Author):

•The authors have addressed all comments extensively. Some answers are not fully convincing (e.g., conditions for the shell to stick vs. permanency of the water film). But, this is not a showstopper from my perspective.

Permanency of the water film during the few hours of an experiment reflects the initial depletion in methane and the slowness of diffusion in either direction through the hydrate shell. Adhesion to the substrate vs. cohesion of the hydrate depends on several factors, including: the properties of the hydrate, those of the surface (*cf.* Ref. 21, Jung & Santamarina, *Geochem., Geophys, Geosys.* 2011), the direction of the effort- mostly traction down the length of the halo, but also shearing at the tip where it adheres to the glass- and the geometrical sections involved, which may provide a mechanical advantage one way or the other, here unfavourable to the very thin halo.

Without adhesion to the glass, it is hard to see how the whole hydrate shell would not move on lowering the pressure on its water side. The observation is that the whole halo stretches proportionally to the distance from the tip which moves less than is detectable, if at all.

•In fact, the manuscript does provide unique data (in the right direction), and advances the understanding of crystal granularity on hydrate properties.

•Hydrates are very difficult to test and characterize. The measurement technique developed by the authors is unprecedented. But, underlying assumptions in indirect/coupled measurements add uncertainty (proper recognition will not be a weakness).

Added “In relation to earlier work, the data appear to be the first estimation of an elastic modulus of methane hydrate by a traction test at micron scale, albeit contactless, unconventional and requiring a measure of theory and assumptions justified as far as possible here. With these restrictions, the present method gives access to the non-linear regime beyond the small displacements probed by acoustic or Brillouin scattering experiments.” (Paragraph starting bottom **p. 14**)

- Hydrates in sediments: Tensile strength is one component. Bonding strength to mineral surfaces and pore-filling-dependent dilation can be equally or more important.

Added the proviso “The mechanical properties of such complicated systems as hydrate bearing sediments depend on many factors including the properties of the hydrate itself, those of the sediment, the hydrate pore habit, the adhesion between hydrate and sediment vs. cohesion of the hydrate.” (l. 1 p. 3)

- Time scale: Nice explanation based on depleted methane; still, there is a time scale for hydrate formation (gas dissolution, clathrate formation, etc).

Indeed! Now mentioned in the caveats of laboratory studies of gas hydrates. (l. 8, p. 3)

- Pg. 16: the measurement is neither direct nor the first. Acoustic properties known for decades.

By direct we meant “by a traction test”. We agree it takes a dose of theory to relate the present thermal excursions to the stress; but there is also a dose of theory between acoustic and Brillouin scattering measurements and an elastic modulus.

Reworded more precisely. (*cf.* above, para. Starting bottom p. 14)

- Text would benefit from edits by an English reader who is not an author. Some highlighted lines in the pdf attached to these comments. While we value the authors thorough response, it is good to keep some proportionality between the length of the text and the length of lessons learned.

Done.

- Page 1: State carefully. Distinguish the properties of the hydrate mass from the properties of the hydrate bearing sediment.

Abstract reworked. (p. 1)

- Text related to test procedure on pg 15 belongs earlier (near pg. 4)

This paragraph, intended as a (helpful?) recap for the reader was kept short, but we agree was indeed redundant with material on p. 4. Removed in the 2nd revision.